

# miR-378 associated with proliferation, migration and apoptosis properties in A549 cells and targeted NPNT in COPD

Guoqing Qian[1,2,3], Qi Liao[4], Guoxiang Li[2] and Fengying Yin[2]

[1] Department of Infectious Diseases, Ningbo Hospital of Zhejiang University, Zhejiang University, Ningbo, Zhejiang, China
[2] Department of Infectious Diseases, Ningbo First Hospital, Ningbo University, Ningbo, Zhejiang, China
[3] Division of Respiratory Medicine, National Institute for Health Research, Nottingham Biomedical Research Centre, University of Nottingham, Nottingham, Nottinghamshire, United Kingdom
[4] Department of Preventative Medicine, Zhejiang Provincial Key Laboratory of Pathological and Physiological Technology, School of Medicine, Ningbo University, Ningbo, Zhejiang, China

## ABSTRACT

**Background:** microRNAs contribute to the development and progression of chronic obstructive pulmonary disease (COPD). However, the underlying molecular mechanisms are largely unclear. The goal of this study was to investigate the roles of miR-378 in alveolar epithelial type II cells and identify molecular mechanisms which contribute to the pathogenesis of COPD.

**Materials and methods:** Human alveolar epithelial (A549) cells were cultured in Dulbecco's Modified Eagle Medium. Cell proliferation was studied by using a cell counting kit-8 (CCK-8) and colony formation assays. Cell apoptosis and cell cycle were analyzed by flow cytometry and wound healing and Transwell were used to analyze the cell migration and. We performed bioinformatics analysis including target gene prediction, gene ontology (GO), Kyoto Encyclopedia of Genes and Genome (KEGG) pathway enrichment and construction of protein-protein interaction (PPI) network. The expression of miR-378 and NPNT from publically available expression microarray of COPD lung tissues was analyzed.

**Results:** Overexpression of miR-378 significantly increases cell proliferation, migration, and suppress apoptosis. GO analysis demonstrated that the miR-378 involved in transcription, vascular endothelial growth factor receptor signaling pathway, phosphatidylinositol 3-kinase signaling, cell migration, blood coagulation, cell shape, protein stabilization and phosphorylation. Pathway enrichment showed that the 1,629 target genes of miR-378 were associated with mTOR, ErbB, TGF-β, MAPK, and FoxO signaling pathways. Notably, miR-378 directly targets Nephronectin in A549 cells, and miR-378 was upregulated while NPNT was downregulated in COPD lung tissue samples.

**Conclusions:** These findings suggest that miR-378 can regulate the proliferation, migration, and apoptosis of A549 cells and target NPNT. miR-378 increased in COPD lung tissues while NPNT decreased, and might prove a potential target for novel drug therapy.

**Subjects** Internal Medicine, Respiratory Medicine, Medical Genetics

Corresponding authors
Guoqing Qian,
bill.qian@outlook.com
Fengying Yin, yfyno1@foxmail.com

**Keywords** NPNT, Nephronectin, Emphysema, COPD, Chronic obstructive pulmonary disease, microRNA, miR-378, microRNA-378, Alveolar epithelium, Alveolar epithelial cells

## INTRODUCTION

Chronic obstructive pulmonary disease (COPD) is the third leading cause of death worldwide (*Lozano et al., 2012*). Pulmonary emphysema accounts for the majority of COPD cases and no effective treatment is currently available. Emphysema causes alveolar epithelial type II (AEII) cell death, permanent destruction of distal airspaces, and chronic inflammation (*Jeffery, 1998*). AEII cells play a vital role in generating surfactant, maintaining alveolar fluid homeostasis, and healing epithelium (*Lin et al., 2019*). Cigarette smoking, the main risk factor for emphysema, triggers alveolar epithelial (A549) cell dysfunction and/or death (*Vij et al., 2018*). However, emphysema's underlying mechanism is not fully understood (*Mannino, 2015*).

Non-coding RNAs (~22 nucleotides long), also known as microRNAs (miRNAs or miRs), are short, single-stranded, and commonly detected in animals (*De Smet et al., 2015*). They regulate a wide range of biological behaviors, including cell proliferation, apoptosis, differentiation, migration, and invasion. They bind to the 3′-untranslated region (UTR) of target mRNAs and mark them for degradation to inhibit their translation (*Christenson et al., 2013*). A number of researchers have reported that miRNAs play an important regulatory role in the biological processes associated with COPD pathogenesis (*De Smet et al., 2015*). miRNAs are also involved in lung development and airway epithelium differentiation, and a previous study found 25 upregulated miRNAs and nine downregulated miRNAs in the small airway epitheliums of patients with a history of smoking (*Wang et al., 2015*). Conversely, a different study recognized that miR-218-5p is a protective factor as it was significantly downregulated in COPD patients (*Conickx et al., 2017*).

Recently, *Martinez-Anton et al. (2013)* found that miR-378 was 8.74 times more upregulated in human bronchial differentiating epithelial cells than in basal epithelial cells. miR-378 was highly expressed in the A549 cell line and was identified as a key miRNA, controlling invasion and migration *via* epithelial mesenchymal transition (EMT) (*San Ho, Noor & Nagoor, 2018*; *San Ho et al., 2014*). Furthermore, canonical correlation analysis showed that miR-378 targets *CDKN1A* in COPD (*Hua et al., 2018*), suggesting that it is involved in epithelium-related pulmonary diseases, particularly COPD. *Kahai et al. (2009)* reported that miR-378 overexpression causes nephronectin (NPNT) downregulation and inhibits osteoblast cell differentiation. NPNT, an important ligand for integrin α8β1, is a novel extracellular matrix linked to numerous biological functions including cell proliferation, adhesion, and differentiation. It is expressed in a variety of organs and tissues, such as the lung, kidney, bones, and liver (*Brandenberger et al., 2001*; *Sun et al., 2018*; *Wain et al., 2015*). Previous studies have found that NPNT is highly expressed in fetal and adult lung tissue as well as alveolar epithelial cells, and is also involved in lung development and function (*Hancock et al., 2010*; *Miller et al., 2016*; *Qian et al., 2016*). However, the role of miR-378 in COPD has not yet been explored.

In this study, our objectives were to investigate the role of miR-378 in A549 cells and to identify potential molecular mechanisms. We found that miR-378 promotes A549 cell proliferation and migration, suppresses cell apoptosis, directly targets NPNT, and upregulates in COPD lung tissues while NPNT downregulates.

## MATERIALS AND METHODS

### Cell culture

We cultured A549 cells (Genechem, Shanghai, China) in Dulbecco's modified Eagle medium (DMEM) with 10% fetal bovine serum (FBS), 100 U/ml penicillin, and 100 mg/ml streptomycin. The cell line was cultured in a cell incubator with 5% $CO_2$ and saturated humidity at 37 °C. We used a cell counting kit-8 (CCK-8) (Sigma, St. Louis, MO, USA) to determine cell viability and measured cell absorbance at a 450 nm wavelength (*Qian et al., 2013*).

### Lentiviral vector production, titration, and transduction

We obtained the pre-hsa-miR-378 sequence from human genomic DNA. The sequence was inserted into a pGCSIL-GFP lentivirus vector with T4 DNA ligase, and the vector was then transformed into competent *Escherichia coli*. HEK293T cells were co-transfected with vector plasmids to produce lentiviral vectors. We plated $3–5 \times 10^4$ A549 cells in a six-well plate to reach 15–30% confluence. They were infected at a multiplicity of infection (MOI) of 20 with polybrene (5 µg/ml), and the medium was changed after 12 h. We confirmed the efficiency of miR-378 infection after 72 h using a fluorescence microscope.

### RNA extraction and quantitative real-time polymerase chain reaction (qRT-PCR) analysis

TRIzol reagent was used to extract A549's total RNA, which was then stored at –80 °C before qRT-PCR analysis. We followed the manufacturer's instructions provided with the reverse transcription kit and primers (RiboBio, Guangzhou, China) to extract total RNA and convert it into cDNA. The qRT-PCR system was set according to the SYBR Green protocol. We calculated the cycle threshold (Ct) value from each reaction tube, and the miR-378 relative expression quantity using the $2^{-\Delta\Delta Ct}$ method.

### Cell proliferation assay

We detected cell proliferation using the previously described CCK-8 assay (*Qian et al., 2013*). A549 cells were seeded into different 96-well plates (2,000 cells/well) and cultured for 5 days. Cell viability was measured using the manufacturer's protocol after adding 10 µl/well of CCK-8 solution.

### Colony formation assay

We used a colony formation assay to detect A549 cell growth. Cells were seeded in six-well plates at a density of 400–1,000 cells/well. The culture medium was changed every 3 days for 14 days, and the colonies were fixed in 4% paraformaldehyde for 30–60 min periods. After fixing, we stained the colonies with 0.5% crystal violet for 10 min on ice and washed each well three times with phosphate-buffered saline.

## Cell cycle analysis

We cultured the A549 cells in a six-well plate and transfected them with miR-378-GFP-LV lentivirus for 4 days to prepare for cell cycle detection. A549 cells were trypsinized and fixed with 70% ethanol at 4 °C for 1 h. We then stained them for 1 h with 500 µl propidium iodide (PI) staining solution containing RNase (*Krishan, 1975*). We used a Guava easyCyte HT flow cytometer (Millipore, Burlington, MA, USA) to detect the cell cycles (300–800 cell/s).

## Cell apoptosis analysis

We cultured the cells in the six-well plate as previously described, and prepared them for cell apoptosis analysis. A549 cells were digested by EDTA-free trypsin and washed in a 4 °C D-Hanks solution (pH = 7.2–7.4). We added $1 \times$ binding buffer, re-suspended the cells twice, and adjusted the cell concentration to approximately $1 \times 10^6$ cells/mL. After centrifugation, the cells were stained with 10 µL Annexin V-APC in darkness at room temperature for 15 min. We then added 400 µL binding buffer to the reaction tubes and used the flow cytometer to detect cell apoptosis (*Pan et al., 2014*).

## Wound healing migration assay

We performed wound-healing migration assay using the 96 wounding replicators. The cells were seeded into each well ($3 \times 10^4$ cells/mL) and incubated at 37 °C, 5% $CO_2$ for 24 h. We used sterile tweezers to scratch the attached cells and washed them with DMEM three times. A microscope was used to detect cell migration (*Espada et al., 2015*).

## Transwell migration assay

We detected cell migration in miR-378-GFP-LV lentivirus-infected group and control group using Transwell assays (Corning, NY, USA) (*Stagg et al., 2013*). Cells were trypsinized, re-suspended, and added in triplicate wells ($1 \times 10^5$ cells/mL). A gentian violet assay was used to quantify and calculate the number of migratory cells per field after 24 h.

## miR-378 target gene prediction and functional analysis

miR-378 target genes were predicted using four online databases: miRDB (http://www.mirdb.org/miRDB), TargetScan (http://www.targetscan.org), miRTarBase (https://bio.tools/mirtarbase), and PicTar (https://tools4mirs.org/software/target_prediction/pictar/). We analyzed the gene ontology (GO) and Kyoto Encyclopedia of Genes and Genome (KEGG) pathways using the Database for Annotation, Visualization, and Integrated Discovery (DAVID) (https://david.ncifcrf.gov/). The inflammation pathway results were entered into Cytoscape (Version 3.7.1; https://cytoscape.org/) to the visualize interaction network. Cytoscape is a powerful software used to integrate biomolecular interaction networks (*Shannon et al., 2003*), visualize protein-protein interaction (PPI), and identify hub genes among potential targets.

## NPNT 3′UTR cloning and luciferase assay

We transfected A549 cells with a NPNT 3′UTR plasmid. Luminescence was assayed after 72 h using the Dual-Luciferase Reporter Assay System (Promega, Madison, WI, USA)

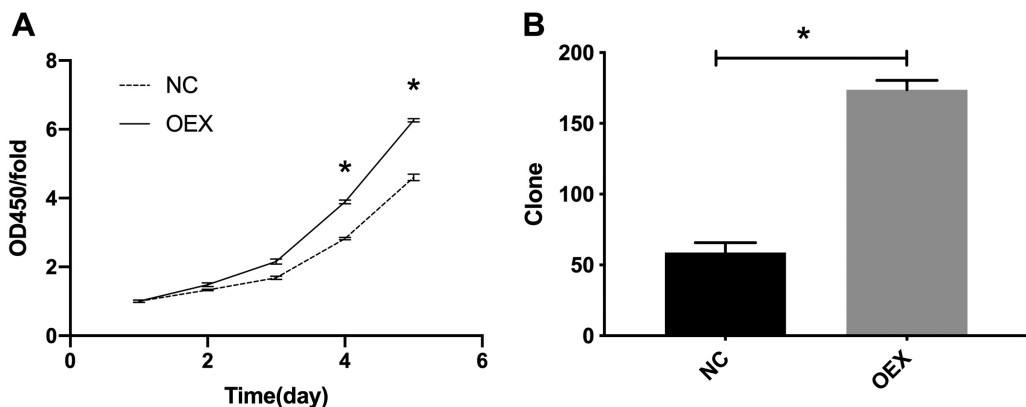

**Figure 1** **Proliferation was analyzed by CCK-8 and colony formation assay.** (A) Overexpression with miR-378 enhanced cells Proliferation. (B) Overexpression with miR-378 promoted cells Colony formation efficiency ($^*p < 0.05$, $n = 3$). CCK-8, Cell Counting Kit-8; NC, control; OEX, overexpression with miR-378.                                               

according to the manufacturer's protocol. We corrected the assay results for background, normalized the values to the control, and expressed them as percentages.

## miRNA and mRNA profiling by array

Prior studies have collected human lung tissue from smokers with COPD ($n = 26$) and without COPD ($n = 9$) to assess for miRNA and mRNA expression by microarray. The data from these studies are available at the National Center for Biotechnology Information's (NCBI) Gene Expression Omnibus (accession number, GSE38974) (*Ezzie et al., 2012*). We compared the miR-378 and NPNT expression levels between the groups with and without COPD by looking at the different Global Initiative for Obstructive Lung Disease (GOLD) stages.

## Statistical analysis

All experiments were independently performed three times. We collected the data in triplicate and expressed it as mean ± SD. Statistical significance was determined by a Mann-Whitney U-test. All statistical analyses were performed using the GraphPad Prism 7 (GraphPad Software, San Diego, California, USA). $p$ values <0.05 were considered significant. An adjusted $p$ value <0.05 and fold change ≥1.5 or ≤−1.5 was considered a statistically significant threshold in our microarray data analysis.

## RESULTS

### miR-378 enhances A549 cell proliferation

The CCK-8 proliferation expression results are summarized in Fig. 1A. The A549 cells' viability in the miR-378 overexpression group was significantly higher than in the control group ($p < 0.001$). We also assessed cell proliferation using colony formation assays after overexpressing miR-378 in A549. The number of cells in the overexpression group was considerably higher (59 ± 7 *vs.* 174 ± 7, $p < 0.001$) (Fig. 1B) than the number in the control group.
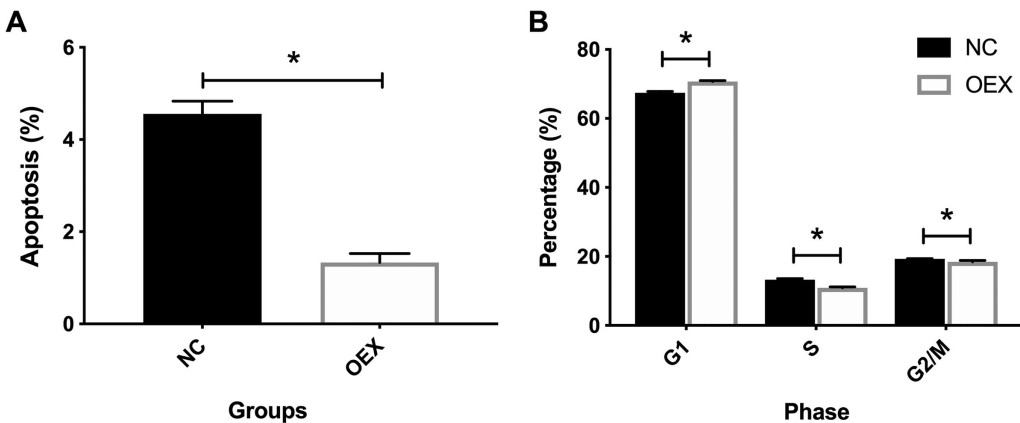

**Figure 2 Effect of overexpression with miR-378 on cell apoptosis and cell cycle distribution.**
(A) Apoptosis percentage of cells in control and overexpression with miR-378 group. (B) Cell cycle
distribution of control, overexpression with miR378 groups after 96 h ($^*p < 0.05$, $n = 3$).

## miR-378 suppresses A549 cell apoptosis and alters cell cycle proportion

Additionally, we identified the effects of miR-378 on A549 cell cycle and apoptosis.
As shown in Fig. 2A, miR-378 overexpression significantly suppressed A549's apoptosis
rate. Compared with the control group, miR-378 overexpression increased the proportion
of cells in the G1 stage, and decreased the proportion of cells in the S and G1/M stages
(Fig. 2B). Our results indicate that miR-378 suppresses cell apoptosis and elevates cell
proliferation by regulating the cell cycle.

## miR-378 promotes A549 cell migration and invasive ability

To assess A549 migration, we initiated stable miR-378 overexpression in an A549 cell line
(miR-378-GFP-LV) using pGC-SIL-GFP lentivirus vector transduction. Compared with
the control vector (control-GFP-LV) group, we observed significantly increased miR-378-
GFP-LV expression (Fig. 3). Additionally, when assessing migration using wound-healing
migration assays, we noticed a significant increase in the miR-378-GFP-LV group
compared with the control-GFP-LV group (Fig. 4). These results suggest that miR-378
upregulates the migration rate of A549 cells.

## Identifying miR-378 target genes

A total of 817 target genes were found in miRDB, 213 target genes in TargetScan, 675
target genes in miRTarBase, and 127 target genes in PicTar. In the expression profiling
datasets, we identified 1,629 target genes from at least one of four target prediction
algorithms (Table S1).

## GO and KEGG pathway functional analysis

To understand the probable biological functions and signaling pathways of miR-378 target
genes, we uploaded the 1,629 target genes to DAVID to perform GO and KEGG pathways
analyses. The GO biological processes include transcription, the vascular endothelial

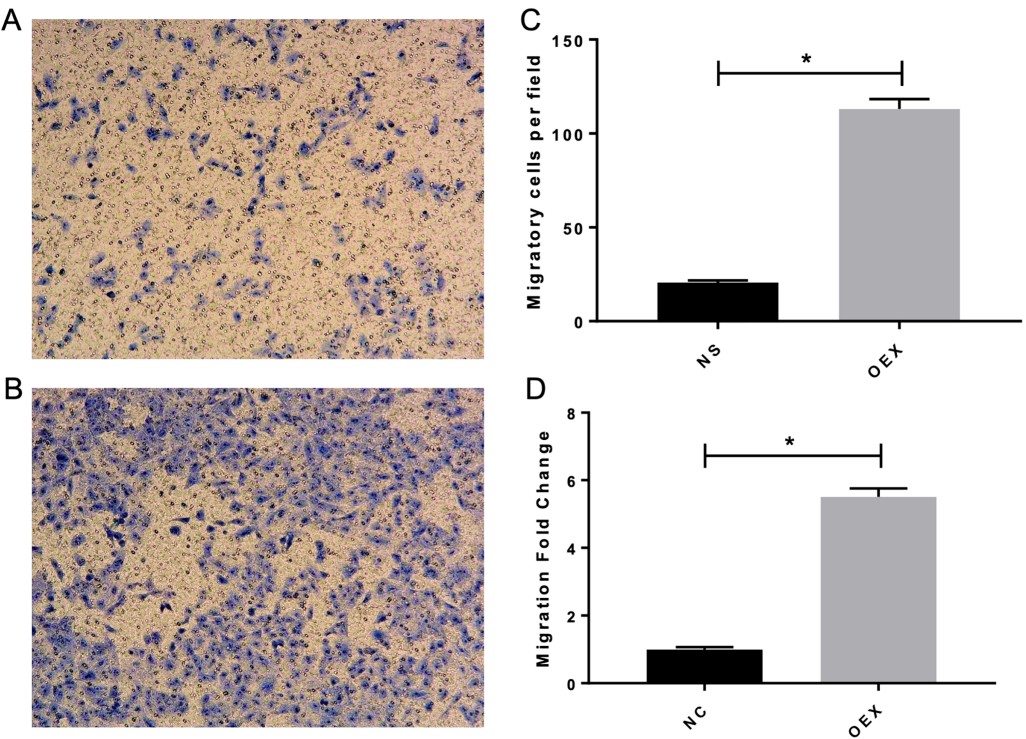

**Figure 3 Overexpression with miR-378 enhanced the migratory ability on alveolar epithelium type II cells.** The transwell assay demonstrated overexpression with miR-378 (B) enhanced cell migration ability compared to control group (A). The images of migratory of cells were taken by phase contrast microscopy under 100×. The data of the cell migration were the average number of cells (C) and cells number fold change (D) (*$p < 0.05$, $n = 3$).

growth factor receptor signaling pathway, phosphatidylinositol 3-kinase signaling, cell migration, blood coagulation, cell shape regulation, protein stabilization, and protein phosphorylation (Fig. 5, Table S2). The target genes enriched in KEGG, mainly the inflammation pathways, were imported into Cytoscape to visualize the interaction network. The results are summarized in Fig. 6.

## miR-378 targets NPNT 3′UTR

A previous study reported that miR-378 targets and regulates NPNT during stable miR-378 transfection of MC3T3-E1 cells with inhibited cell differentiation (*Kahai et al., 2009*). We also reached this conclusion based on our database predictions. The entire wild-type NPNT 3′UTR and mutant 3′UTR were cloned downstream of the luciferase gene and assayed in A549 cells, respectively. As shown in Fig. 7, there was a significant difference between the 3′UTR-NC and 3′UTR ($p < 0.05$). These results suggest that NPNT is directly targeted by miR-378 in A549, the human basal alveolar cell line.

## miR-378 is upregulated while NPNT is downregulated in COPD

We used publicly available expression microarray data to assess miR-378 expression in lung tissues taken from 26 COPD patients with a history of smoking and nine individuals with a history of smoking but no evidence of obstructive lung disease. miR-378 expression

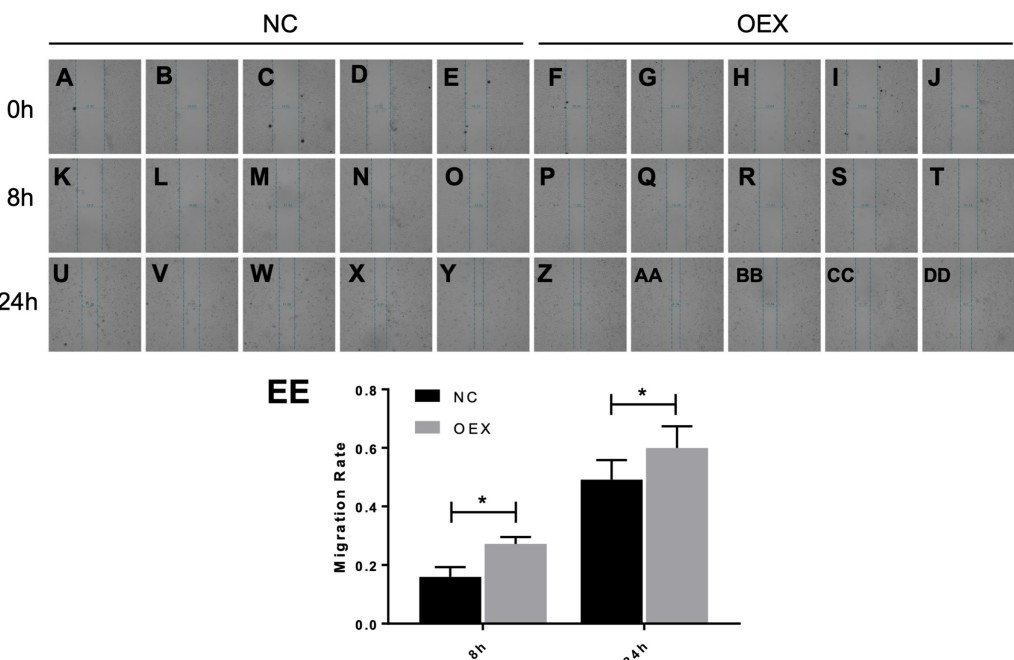

**Figure 4 Effect of overexpression with miR-378 on the migration of alveolar epithelial cells.** Scratch wounds were created in cell monolayers of human alveolar epithelial cells (A) by a sterile tweezers, as well as the data of migration rate (B) ($*p < 0.05$, $n = 3$).

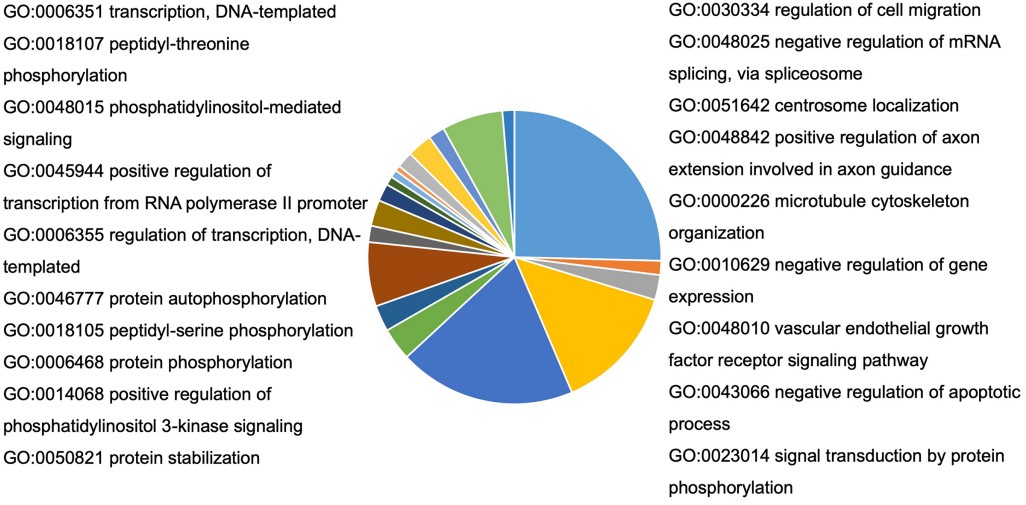

GO:0006351 transcription, DNA-templated
GO:0018107 peptidyl-threonine phosphorylation
GO:0048015 phosphatidylinositol-mediated signaling
GO:0045944 positive regulation of transcription from RNA polymerase II promoter
GO:0006355 regulation of transcription, DNA-templated
GO:0046777 protein autophosphorylation
GO:0018105 peptidyl-serine phosphorylation
GO:0006468 protein phosphorylation
GO:0014068 positive regulation of phosphatidylinositol 3-kinase signaling
GO:0050821 protein stabilization

GO:0030334 regulation of cell migration
GO:0048025 negative regulation of mRNA splicing, via spliceosome
GO:0051642 centrosome localization
GO:0048842 positive regulation of axon extension involved in axon guidance
GO:0000226 microtubule cytoskeleton organization
GO:0010629 negative regulation of gene expression
GO:0048010 vascular endothelial growth factor receptor signaling pathway
GO:0043066 negative regulation of apoptotic process
GO:0023014 signal transduction by protein phosphorylation

**Figure 5 Bioinformatic analysis of predicted target genes of miR-378.** GO biological process (BP) of target genes. A total of 27 pathways were selected by $p < 0.05$ and FDR < 5%. BP, biological process; GO, gene ontology.

increased more in the COPD patients than in the smokers with no history of COPD (adjusted $p$ value < 0.05 and logFC = 0.62, Fig. 8A). On the other hand, NPNT decreased more in individuals with COPD compared with smokers who had no history of COPD (adjusted $p$ value < 0.05 and logFC = −0.97, Fig. 8B).

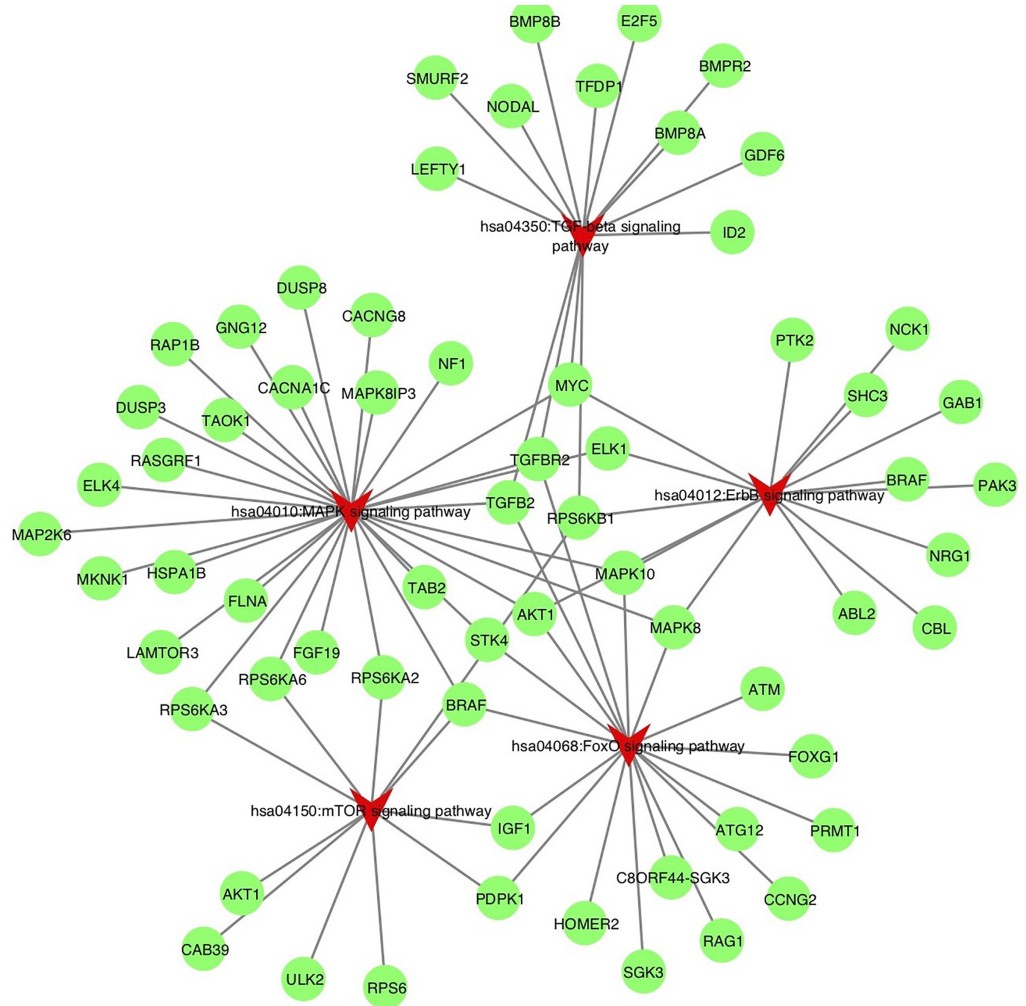

**Figure 6 Target genes implicate in inflammation process potentially altered by miR-378.** The enriched KEGG pathways of the target genes involved in inflammation process, including hsa04012: ErbB signaling pathway, hsa04350: TGF-beta signaling pathway, hsa04150: mTOR signaling pathway, hsa04010: MAPK signaling pathway, hsa04068: FoxO signaling pathway. The pathways are highlighted by red. TGF-beta, transforming growth factor-beta; MAPK, mitogen-activated protein kinase; FoxO, Forhead box O, ErbB, also called EGFR, epidermal growth factor receptor.

## DISCUSSION

Our results shows that miR-378 overexpression in A549 cells promotes cell proliferation, migration, and invasion, and suppresses cell apoptosis. Furthermore, we have shown that miR-378 increases in COPD and is involved in the development and progression of COPD *via* NPNT regulation.

COPD is a major cause of morbidity and mortality worldwide that comprises a range of diseases, including bronchitis and emphysema. Pulmonary emphysema, seen in only 3% of non-smokers, develops in approximately 30–50% of cigarette smokers (*Auerbach et al., 1972*; *Manichaikul et al., 2014*). Many studies have reported on the genetic and environmental factors involved in the development of emphysema, and the role of

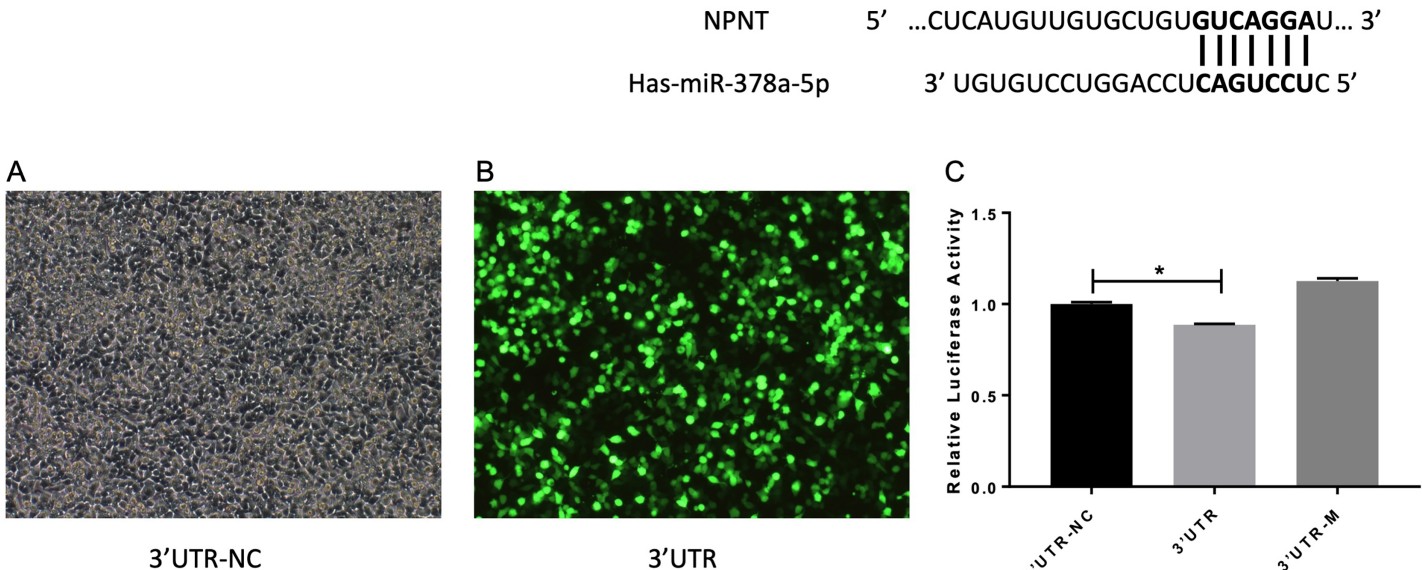

Figure 7 (A–C) miR-378 target NPNT in human alveolar epithelial cells. Prediction and identification of miR-378 target 3′UTR of NPNT by luciferase (*$p < 0.05$). NPNT, Nephronectin; UTR, untranslated regions.

miRNAs in the development and progression of COPD (*Liu et al., 2018*; *Mannino, 2015*). A previous study by *Francis et al. (2014)* showed the downregulation of miR-34c, miR-34b, miR-149, miR-133a, and miR-133b and involvement of miR-34c in the severity of emphysema. *Christenson et al. (2013)* reported that regional emphysema altered the expression of 63 miRNAs, and that target gene functional pathways were involved in inflammation as well as extracellular matrix and tissue repair.

miR-378's role in COPD is largely unknown. miR-378, located at chromosome 5q32, is associated with lung cancer, osteoblast differentiation, muscle development, and angiogenesis (*Ji et al., 2018*; *Kahai et al., 2009*). Cytokines such as transforming growth factor (TGF)-β1 can also be upregulated by miR-378 (*Ong et al., 2017*). Although miR-378 expression is not significantly different between mild and moderate emphysema (*Christenson et al., 2013*; *Francis et al., 2014*), miR-378 is highly expressed in A549 cells and is involved in COPD (*Hua et al., 2018*; *San Ho, Noor & Nagoor, 2018*). Our results are consistent with previous studies by indicating that miR-378 can promote epithelial cell proliferation and suppress cell apoptosis by altering the cell cycle (*Ji et al., 2018*). Additionally, miR-378 overexpression can promote the rate of epithelial cell migration and invasion. Overall, these results suggest that miR-378 plays an important role in epithelial cells and should be further studied.

To further identify miR-378's function, we performed GO and KEGG pathway analyses to predict the biological function and pathways of target genes. The results suggest that miR-378 targets play critical roles in inflammation, cell migration, transcription, and protein stabilization. Enriched pathways include the phosphatidylinositol 3-kinase (PI3K), mTOR, vascular endothelial growth factor receptor, ErbB, TGF-β, mitogen-activated protein kinase (MAPK), and Forkhead box O (FoxO) signaling pathways. There is increasing evidence that PI3K-mTOR is activated in the lungs and cells of COPD patients

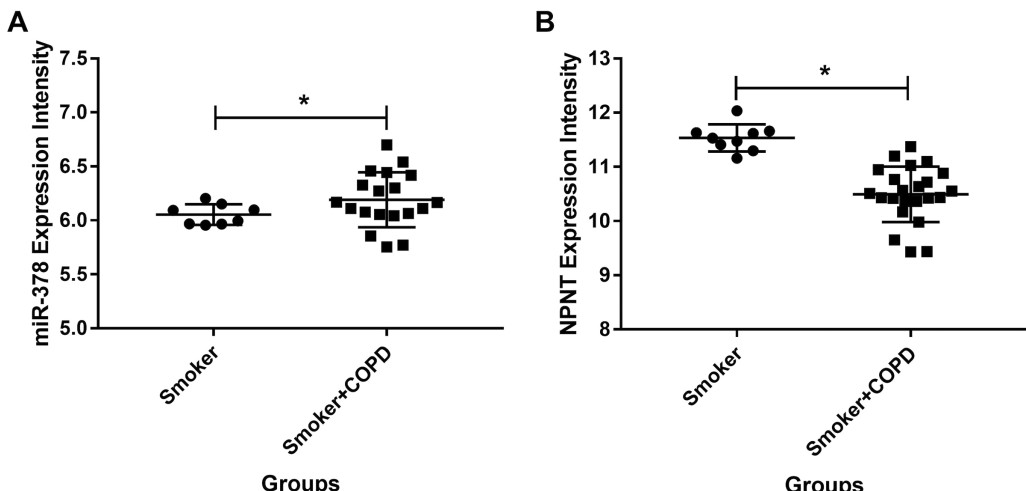

**Figure 8 The miR-378 and NPNT expression intensity in smoker and smoker with COPD lung tissue samples.** The miR-378 upregulated and NPNT downregulated in lung tissue in GSE38974, including 26 smokers with COPD and nine smokers without COPD (*adjusted $p$ value < 0.05). GSE, series of the Gene Expression Omnibus.                     

(*Barnes, 2017*). Studies have also identified vascular endothelial growth factor (VEGF) as a biomarker of oxidative stress and COPD severity (*Byers, 2019*). There is also evidence implicating the ErbB (also called epidermal growth factor receptor, or EGFR) pathway in pulmonary emphysema (*Su, Luo & Yang, 2018*; *Ng-Blichfeldt et al., 2019*). Smoking promotes alveolar epithelial cell dysfunction by activating the MAPK pathway (*Zeglinski et al., 2019*). The evidence strongly suggests that miR-378 may contribute to airway inflammation and remodeling *via* the enriched pathways mentioned above.

Previous studies have suggested that miR-378 directly targets genes, including osteoblast NPNT (*Kahai et al., 2009*), Forkhead box G1 (FOXG1) in lung cancer (*Ji et al., 2018*), and bone morphogenetic protein 2 (BMP2) in C2C12 cell lines (*Yang, Luo & Wang, 2019*). Our previously published study reported that NPNT is highly expressed in lung tissue, particularly in the alveolar cells which play an important role in lung development (*Hancock et al., 2010*; *Qian et al., 2016*; *Qian, Yang & Shi, 2019*; *Wain et al., 2015*). We also confirmed that miR-378 directly targets NPNT in the epithelial cells and is upregulated (while NPNT is downregulated) in the lungs of COPD patients. This suggest that miR-378 is involved in the development and progression of COPD in people with a history of smoking. Further studies need to be performed to confirm the role and function of miR-378 in lung development.

Our study had some limitations that should be taken into account when interpreting the results. First, we used the lung cancer cell line A549. Future experiments should be performed on BEAS-2B or primary human bronchial epithelial cells. Second, we did not collect samples nor confirm miR-378 expression using qRT-PCR in lung tissue from individuals with COPD or smokers without airflow limitations. Third, using a luciferase report assay and qRT-PCR or Western blot/immunohistochemistry might better assess the underlying biological mechanism of the target genes and find more potential targets.

## CONCLUSIONS

Our results show that miR-378 promotes proliferation and migration and suppresses apoptosis in A549 cells. We have also demonstrated that miR-378 directly targets NPNT and is upregulated in the lung tissue of COPD patients, while NPNT is downregulated. Therefore, miR-378 might play an important role in the pathogenesis of COPD.

### Funding

This work was supported by the Zhejiang Provincial Natural Science Foundation of China under Grant No. Q17H010001, the Ningbo City Natural Science Foundation of China under Grant No. 2017A610246 and 202003N4019, the Projects of Medical and Health Technology Development Program in Zhejiang Province under grant No. 2017KY573. The funders had no role in study design, data collection and analysis, decision to publish, or preparation of the manuscript.

### Grant Disclosures

The following grant information was disclosed by the authors:
Zhejiang Provincial Natural Science Foundation of China: Q17H010001.
Ningbo City Natural Science Foundation of China: 2017A610246, 202003N4019.
Projects of Medical and Health Technology Development Program: 2017KY573.

### Competing Interests

The authors declare that they have no competing interests.

### Author Contributions

- Guoqing Qian conceived and designed the experiments, performed the experiments, analyzed the data, prepared figures and/or tables, authored or reviewed drafts of the article, and approved the final draft.
- Qi Liao performed the experiments, analyzed the data, prepared figures and/or tables, authored or reviewed drafts of the article, and approved the final draft.
- Guoxiang Li conceived and designed the experiments, authored or reviewed drafts of the article, and approved the final draft.
- Fengying Yin conceived and designed the experiments, authored or reviewed drafts of the article, and approved the final draft.

### Data Availability

  The raw measurements are available in the Supplemental Files.

### Supplemental Information

Supplemental information for this article can be found online at http://dx.doi.org/10.7717/peerj.14062#supplemental-information.

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
