# Peer review of "miR-378 associated with proliferation, migration and apoptosis properties in A549 cells and targeted NPNT in COPD"

_PeerJ, doi:10.7717/peerj.14062_

## Round 0.1 · original submission · Major Revisions

The manuscript was examined by three reviewers, two of whom suggested a 'reject' decision. But I hope that you will be able to appropriately address the reviewers' concerns with a major revision to the manuscript. All the concerns of the reviewers are important, including those related to English and presentation.

·

Basic reporting

The English is professional, but could be improved in some cases to add clarity to presentation of findings.
The results presented in the paper do not support the hypothesis given. The authors aim to resolve the importance of miR-387 in chronic obstructive lung disease (COPD), but use a cancer cell line, A549, in their experiments. This is the main flaw in the design of the study. The results that can be interpreted to be relevant to COPD and to support the aim of the study are the in silico analysis of targets for miR-378 (which in itself is not focused enough and their presentation could be improved) and the analysis of publicly available data on the levels of expression of miR-378 and nephronectin in COPD patients. These are not enough to fulfill the proposed aim. The way to improve the study and provide relevant results to support the hypothesis could be to do the experiments on some non-cancer lung epithelial cell line or primary cell line from COPD patients (which are available commercially).

Experimental design

This manuscript represents original research and falls into the Aims and Scope of the journal. However, model used for the experiments was not appropriate, since lung cancer cell line - A549 was used in all the experiments and the results were discussed considering the findings as valid for COPD.
Additionally, Material and methods section could be significantly improved. The information on the setup of experiments is lacking, not enough information is given to replicate the experiments.
There is no information on the concentration of the reagents used, only volume added. The type of the FACS machine used should be given. The details about the statistical analysis of cell cycle experiment should be given. There is no information in the text if all the experiments were done each time in triplicate (technical repeats).
The data about the analysis of overexpression of miR-387 using RT-PCR, should be given in supplement. Additionally the authors should give the information about what probe and method was used for this analysis.It is also not clear if the stable transfectants were used in all the experiments.
In Figure 1A, it is better to give error bars, because the lines for different experiments are not visible.
In Figure 5 it is not clear which part of the pie represents which pathway.
In Figure 7 legend the explanation on what A, B and C represent is not given.

Validity of the findings

Although interesting, the results are not applicable to COPD, but more fall into the lung cancer research area. The authors should address this issue before submitting a new paper. There are serious issues in the presentation of the manuscript (some of them are listed in previous chapters) that should be improved. The interpretation of the results should consider the model used in the experiments.

·

Basic reporting

1. The article needs proof reading, the English used is not proficient.
2. The methods are not written in a comprehensive way. Please report the experiments in a way allowing repetition by other scientists.
3. The method section misses important information on procedures performed. The authors report in the result section to have used a 3’-UTR assay analyzing the complete NPNT-UTR for miR-378 cleavage (Line 211 and following). The method section misses the information on primers used, the region spanned, the number of target sites per UTR and mutations introduced to verify sequence specificity. Please include.

Experimental design

1. The authors perform an apoptosis assay on a cancer cell line showing less than 5 % of apoptotic activity. This system doesn’t seem appropriate to study apoptotic responses. The authors should consider including an agent able to introduce apoptosis in A549 cells and then compare miR-378 overexpression to control cells.
2. Lines 193-197. The authors download predicted targets of miR-378 from 4 different online sources. While all prediction tools use different algorithms, the general consensus is to study target genes predicted by more than one algorithm. Please comment on the rationale? Please, also specify target selection criteria in more detail, e.g. highly conserved among mammals or number of consensus nucleotides in seed sequence.
3. The authors perform GO and pathway analysis using the complete list of predicted miR-378 targets. Did the authors verify, that all predicted targets are expressed in ATII cells?

Validity of the findings

1. MiR-378 seems to represent a family of microRNAs (miR-378 a- j). If not all target NPNT, please adjust article to the correct name.
2. Using the four online tools for target prediction, this reviewer was only able to find NPNT among the miR-378 target list at miRTarBase. Please comment and correct own statement!
3. The authors use publicly available data deposited under GSE38974, that summarize gene expression analysis of smokers with and without COPD. The authors detect elevated miR-378 and lower NPNT expression in COPD+ smokers (Fig. 8 A &B). Looking into the deposited data set, the expression profiling summarized COPD data from different Gold stages. Do the authors see a difference among Gold stages for both miR-378 and NPNT expression? Did the authors select a certain Gold stage for further analyses? Please report in method section.

Additional comments

Qian et al investigate the impact of miR-378 on alveolar epithelial apoptosis, response to wound healing and migration, addressing an important question in COPD research. The authors use state of the art techniques comprising a cancer cell line and overexpression methods to assess the impact of miR-378 on epithelial cells. The study presented identifies several interesting aspects, however requires major recisions to improve the quality of the manuscript.
1. Fig.1 A seem to have 4 curves while only 2 groups are annotated. Please correct.
2. The authors perform miR-378 overexpression. Did the authors assess miR-378 and NPNT levels in A549 cells at baseline and after miR-378 over-expression.
3. The differences between groups seem very small and only due to high n, reach statistical significance. Please indicate how many technical and biological replicates have been performed per experiment.

·

Basic reporting

The manuscript entitled “miR-378 is Implicated in Chronic Obstructive Pulmonary Disease and Targets Nephronectin” by Qian et al. shows that overexpression of miR-378 promotes cell proliferation, cell migration and suppresses cell apoptosis in A549 cell line. The authors also describe the function of miR-378 target genes and show that miR-378 targets Nephronectin. Moreover, they show that miR-378 is upregulated in lung tissue samples from smokers with COPD whereas Nephronectin is downregulated.

1. Language requires major improvement. I would recommend a language editing by an English native speaker.
2. Some references are missing, for examples in lines 54, 55, 79, 80, 235, 236.

Experimental design

1. The authors should provide the sequence of the primers used for cloning of pre-miR-378.
2. (OEX) represents cells transfected with lentivirus carrying miR-378 (LV-miR-378). However, it is not clear what does (NC) group stand for. Does it stand for cells transfected with lentivirus carrying empty vector (LV-control) or for un-transfected control cells? In both cases, the control group is missing. Ideally it should be: LV-miR-178 group, LV-control group and un-transfected control group. (Line 98-Lentivirial vector production, titration and transduction).
3. The evaluation of the transfection efficiency, that the authors mention was confirmed by qRT-PCR, should be provided.
4. Fig 2B represents the proportion of cells which are based on the representative images of cell cycle distribution. However, the representative images are not provided. I would suggest to add the original representative images/graphs of cell cycle distribution for each group together with the percentage of cells in various phases (2 figures).

Validity of the findings

1. It has been already shown that the expression of miR-378 in A549 adenocarcinoma epithelial cell line is associated with cell invasion and migration (Chai San Ho et al., 2014, chai San Ho et al., 2018). It has also been shown in another study, which analysed microRNA and mRNA microarray profiles of smokers and COPD patients, that miR-378 is implicated in COPD (Hua Lin et al., 2018-Prediction of microRNA and gene target from an integrated network in chronic obstructive pulmonary disease based on canonical correlation analysis). Please cite these references.
2. The study is partly incomplete as the conclusions are not enough supported by the data. The authors conclude that:
-In line 254 - miR-378 might heal destruction under cigarette exposure or particles. However, in order to draw such conclusions, a proper in vitro COPD model should be used, for example, A549 cells treated with cigarette smoke.
3. The title states “miR-378 is Implicated in Chronic Obstructive Pulmonary Disease and Targets Nephronectin” but the fact that miR-378 targets Nephronectin is known (Müller-Deile et al., 2017, Kahai et al., 2009). Therefore, I would suggest rewriting the title.
4. In Line 168 - “miR-378 enhances cell proliferation in epithelium”, I suggest replacing the word “epithelium” with “A549 cells”. Similarly, for lines 176, 184 and 290. The authors should be careful in drawing the conclusions. They should avoid the generalization of the result as only one cell line was used (line 179- miR-378 overexpression significantly suppressed the apoptosis rate of alveolar epithelium), (line 192- Collectively, miR-378 upregulates the migration rate of alveolar epithelial cells). Therefore, ‘alveolar epithelium’ should be replaced with ‘A549 cells’.

---

## Round 0.2 · Major Revisions

We received the revised version of the manuscript along with your response to the reviewers' comments. Before the revised manuscript is sent to the reviewers, I request you to revise it further to address the following. Please provide responses by changing the manuscript for each of them. I advise you to look at various miRNA papers to understand the depth of method details that are needed in your manuscript.

(1) Details on generation of the hsa-pre-miR-378 lentivirus should be provided:

a. Source of 'genomic DNA' for amplifying hsa-pre-miR-378

b. Primers used

c. Length of amplified product

d. Names of the 'vector plasmids' and their source

e. Transfection method for HEK293T transfection

f. What is the control virus used for the control group? How was it generated?

(2) Details of miR-378 RT-PCR

a. What was the normalizing RNA used for the 2-∂∂Ct method?

b. Please provide its primer sequences in the manuscript.

c. Also provide primer sequences for miR-378. If primer sequences are not known, provide company product numbers in the manuscript.

(3) Details of cell cycle/apoptosis analysis

a. What software was used to analyze flow cytometry data?

b. How was the data analyzed? E.g., gating of dead cells.

c. How were cell cycle values determined from flow cytometry data.

(4) Details of Transwell migration assay

a. How was the gentian violet assay performed. E.g., include cell lysis, colorimetric measurement methods.

(4) Details of NPNT reporter plasmid

a. What was the plasmid vector?

b. How was NPNT amplified (DNA source, primer sequences, etc.) and cloned into the plasmid.

c. What was the length of the cloned region?

d. Provide GenBank accession number of reference NPNT sequence, and location of primers in the sequence (e.g., fron nt 1018 to nt 1068).

e. How was mutagenesis performed?

f. What mutations were generated?

(4) Is there publicly available data from any other human COPD study with both miRNA and mRNA data besides GSE38974? If not, please mention this in Discussion. If there is data from another study, please analyse the data in the same way as you did for GSE38974.

---

## Round 0.3 · accepted · Accept

The revised manuscript was sent to the previous submission's reviewers but unfortunately they did not respond in time. I am therefore making this decision to accept the revised manuscript for publication since it largely addresses the concerns raised in the previous review.